# Protein folding while chaperone bound is dependent on weak interactions

Kevin Wu[1,2], Frederick Stull[1,3,4], Changhan Lee [1,3] & James C.A. Bardwell[1,3]*

It is generally assumed that protein clients fold following their release from chaperones instead of folding while remaining chaperone-bound, in part because binding is assumed to constrain the mobility of bound clients. Previously, we made the surprising observation that the ATP-independent chaperone Spy allows its client protein Im7 to fold into the native state while continuously bound to the chaperone. Spy apparently permits sufficient client mobility to allow folding to occur while chaperone bound. Here, we show that strengthening the interaction between Spy and a recently discovered client SH3 strongly inhibits the ability of the client to fold while chaperone bound. The more tightly Spy binds to its client, the more it slows the folding rate of the bound client. Efficient chaperone-mediated folding while bound appears to represent an evolutionary balance between interactions of sufficient strength to mediate folding and interactions that are too tight, which tend to inhibit folding.

[1] Howard Hughes Medical Institute, University of Michigan, Ann Arbor, MI 48109-1085, USA. [2] Department of Biophysics, University of Michigan, Ann Arbor, MI 48109-1055, USA. [3] Department of Molecular, Cellular, and Developmental Biology, University of Michigan, Ann Arbor, MI 48109-1085, USA. [4]Present address: Department of Chemistry, Western Michigan University, Kalamazoo, MI 49008-5413, USA. *email: jbardwel@umich.edu

Cells rely on a comprehensive chaperone network to mediate protein folding and prevent protein aggregation[1–3]. Although some chaperones utilize cycles of ATP binding and hydrolysis to regulate their chaperone activity, others do not. Most ATP-independent chaperones fail to interact with native proteins, but instead bind tightly to misfolded or partially folded client proteins in order to prevent protein aggregation[4,5]. However, we and others have recently characterized an ATP-independent chaperone called Spy that does not exclusively bind to proteins that are in non-native conformations. Instead, Spy interacts with various folded and unfolded states of its client protein Im7 with similar micromolar affinities[6]. Somewhat surprisingly, Spy allows Im7 to fold while it remains continuously in contact with the surface of the chaperone[6–10]. This chaperone's mechanism does not appear to be unique; there are several other intriguing examples of protein folding while bound to chaperones[11]. However, the chaperone properties that are needed to allow for folding while bound remain unclear. It is also uncertain if Spy generally allows folding while bound or if this is something specific to its well-characterized client Im7. We therefore decided to investigate the folding mechanism of another Spy client that is unrelated to the alpha helical Im7. Here, we find that Spy is also able to refold the β-sheet rich Fyn SH3 protein while it remains bound, suggesting that Spy may be generally able to allow folding while bound. Furthermore, increasing the affinity of client chaperone interactions slows folding—the tighter the interactions, the more folding is slowed. These observations lead us to conclude that chaperone-mediated folding-while-bound represents an evolutionary balance between too tight binding, which can act to inhibit folding, and too loose an association, where the chaperone is unable to significantly affect folding or aggregation.

## Results

**Fyn SH3 interacts with Spy to form a 1:1 complex**. Spy's rapid binding to its well-characterized model substrate Im7 is driven by interactions between negative charges on Im7 and the positive charges that are present on the interior of Spy's cradle-like structure[8–12]. Fyn SH3 is another well-studied folding model protein[13–15] that possesses a net negative charge[16]. NMR experiments have recently shown that native Fyn SH3 can interact with Spy[17]. Using isothermal titration calorimetry (ITC), we confirmed that the Spy dimer binds native SH3 with a dissociation constant ($K_d$) of $50 \pm 6\ \mu M$ (Supplementary Fig. 1a), similar to the ~30 μM affinity that we obtained for the Spy–Im7 interaction[6]. In our previous study, we were able to observe that native Im7 is slowly and partially unfolded upon binding to Spy[6]. In contrast, we did not observe any slow change in SH3 fluorescence that would correspond to a SH3 unfolding event that occurs after Spy binding (Supplementary Fig. 2). This apparent lack of Spy-mediated unfolding of SH3 may be due to the higher thermodynamic stability of native Fyn SH3 ($\Delta G_{UN}$ SH3 = $-16.9 \pm 0.1\ KJ\ mol^{-1}$) relative to that of Im7 ($\Delta G_{UN}$ Im7 = $-13 \pm 2\ KJ\ mol^{-1}$) (Supplementary Table 1).

**Fyn SH3 folds while bound to Spy**. To determine if SH3, like Im7, can fold while bound to Spy, we conducted stopped-flow fluorescence experiments analogous to those we had previously performed with Im7[6]. Like for Im7, the fluorescence of the tryptophan residues in Fyn SH3 are sensitive to Fyn SH3's folding status[15]. Upon diluting Fyn SH3 from urea into buffer in the absence of Spy, the tryptophan fluorescence increased, with a half time of 0.03 s, indicative of rapid refolding (Fig. 1a red trace) consistent with prior reports[14,15]. To verify that SH3 does not aggregate upon dilution from urea, we performed the refolding experiments at several different SH3 concentrations (0.5–8.7 μM).

All SH3 concentrations produced kinetic traces with approximately the same observed rate constant, and the amplitude of the refolding traces increased linearly with the SH3 concentration, indicating that the denaturant-diluted SH3 did not aggregate in our experiments (Supplementary Fig. 3).

As had been observed for Im7, the time required for SH3 refolding increased when the experiments were performed in the presence of increasing concentrations of Spy (Fig. 1a), indicating that Spy, like for Im7, significantly slows the folding of SH3. The kinetic traces for Fyn SH3 folding in the presence of Spy (which is tryptophan-free and therefore does not contribute to these fluorescence changes) could all be described by a single exponential function (Supplementary Fig. 4).

We previously reasoned that if clients need to be released from a chaperone prior to folding, high concentrations of that chaperone should inhibit folding by reducing the amount of unbound, and thus folding-competent, client protein available. Mathematically, mass action dictates that the observed rate constant ($k_{obs}$) for the folding/unfolding reaction should asymptotically approach zero as the chaperone concentration is increased if the client must be released before folding. In contrast, if clients can fold and unfold while continuously bound to a chaperone, $k_{obs}$ should reach a non-zero value at saturating chaperone concentrations. As for Im7, we found that the $k_{obs}$ for the folding of SH3 asymptotically reached a non-zero value at high Spy concentrations (Fig. 1b). This finding strongly suggests that SH3, like Im7, does not have to be released from Spy before folding.

**Spy compacts the unfolded SH3 at the start of folding**. To gain additional insight into whether Spy binding affects the folding pathway of SH3, we conducted stopped-flow intramolecular FRET experiments. Here, we used the two sequential, intrinsic tryptophans (Trp36 and Trp37) present in SH3 as the fluorescence donor and a thionitrobenzoate (TNB) adduct site-specifically labeled via the thiol of a single cysteine mutant (SH3T2C) as the fluorescence acceptor (Supplementary Fig. 5a). The extensive spectral overlap between the tryptophan donor and the TNB fluorophore acceptor makes them ideal for measuring changes in distance in proteins using FRET and thus they have been commonly used for this purpose[18–20]. Multiple clustered tryptophans have been previously used as donor in FRET experiments[18,21]; in our case, since the tryptophans are immediately adjacent, they are tightly clustered, increasing the accuracy of the measurements. In order to validate the spatial accuracy of our introduced FRET pair, we monitored the tryptophan fluorescence changes that occur during the refolding of TNB-labeled SH3 and unlabeled SH3 in the absence of Spy by stopped-flow (Supplementary Fig. 5b). The tryptophan fluorescence of TNB-labeled SH3 decreases during the refolding process, which is consistent with the expected decrease in distance between the Trp-TNB pair when SH3 transitions from an unfolded conformation to the native state (Supplementary Fig. 5b). Notably, $k_{obs}$ for the refolding of TNB-labeled SH3 was similar to that of unlabeled SH3, suggesting that the TNB fluorophore does not alter the folding pathway. We then used the fluorescence intensity of unlabeled ($F_D$) and TNB-labeled SH3 ($F_{DA}$) at each time point during the stopped-flow refolding curve to calculate, using Eq. 1 (See Methods), the FRET efficiency (E) and the apparent distance between the donor and acceptor at each time point during refolding. The resulting kinetic time course shows that the apparent distance between the fluorophores immediately after diluting the denaturant was $22.9 \pm 0.4$ Å (Fig. 2a), indicating that the unfolded state of SH3 is more compact under native conditions than it is in 9.5 M urea ($27.9 \pm 0.1$ Å). Note that the

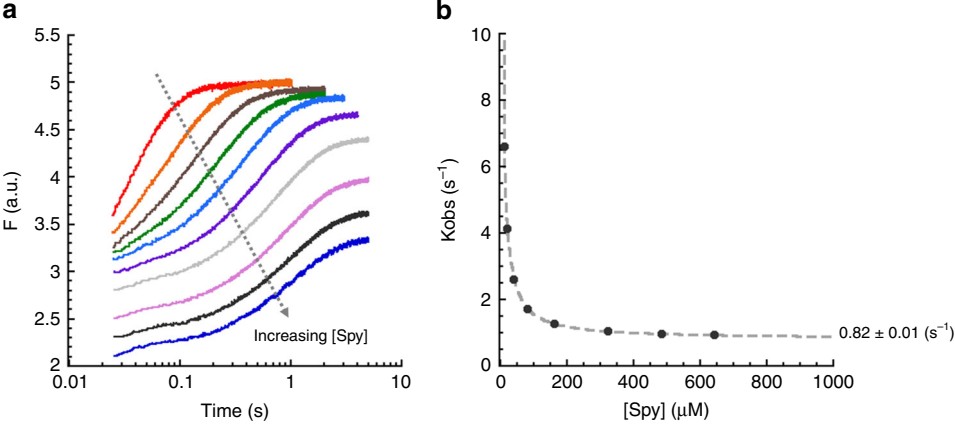

**Fig. 1** Kinetic analysis of SH3 refolding in the presence of Spy. **a** The graph shows the fluorescence traces of SH3 refolding in the presence of various concentrations of Spy (0–643 μM dimer after mixing). All traces could be fit with a single exponential function. The decrease in the fluorescence intensity at high Spy concentrations is due to the inner filter effect caused by tyrosines in Spy. **b** Plot of $k_{obs}$ for SH3 folding as a function of Spy concentration. The $k_{obs}$ hyperbolically decreased as the Spy concentration increased, reaching $0.82 \pm 0.01\,s^{-1}$ at saturation

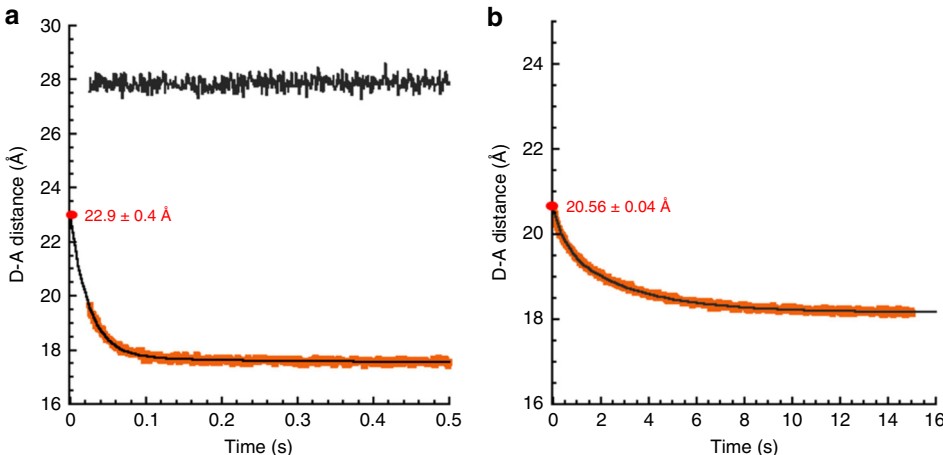

**Fig. 2** Refolding kinetics of SH3 monitored by stopped-flow FRET. **a** The changes in the apparent distance between donor and acceptor (D-A distance), which was calculated from the FRET efficiency (Supplementary Fig. 5d), during SH3 refolding in the absence of Spy (shown as solid orange line). The solid black line shows the D-A distance when SH3 is in buffer containing 9.5 M urea, which was calculated from the FRET efficiency (Supplementary Fig. 5e). **b** The changes in D-A distance for SH3 refolding in the presence of Spy (160.7 μM dimer after mixing), which was calculated from the FRET efficiency (Supplementary Fig. 6b)

distances listed are apparent distances; the relationship these apparent distances have to physical distances is subject to the various considerations that generally apply to FRET measurements[22,23]. During refolding, the SH3 converts from the $22.9 \pm 0.4$ Å unfolded conformation to the native state, with an apparent donor–acceptor distance of $17.55 \pm 0.03$ Å.

We next investigated whether the conformations of TNB-labeled SH3 during refolding are altered when refolded in the presence of Spy and also to determine what the conformation of SH3 is that Spy interacts with. To accomplish this, we conducted the stopped-flow refolding experiment in the presence of Spy and analyzed the apparent donor–acceptor distance of SH3 during the refolding process (Fig. 2b and Supplementary Fig. 6). At time zero during SH3 refolding, the apparent donor–acceptor distance was $20.56 \pm 0.04$ Å in the presence of Spy (Fig. 2b) compared with the $22.9 \pm 0.4$ Å apparent distance in its absence (Fig. 2a). This result suggests that unfolded SH3 is further compacted when bound to Spy, which may be due to the reduced conformational freedom of unfolded SH3 in the bound state. During the subsequent folding of SH3, the apparent donor–acceptor distance further shrinks, and at the end of SH3 refolding in the presence of Spy the

apparent donor–acceptor distance was $18.16 \pm 0.04$ Å (Fig. 2b), very similar to the distance of $17.55 \pm 0.03$ Å found in refolded SH3 in the absence of Spy (Fig. 2a). This observation suggests that SH3 does indeed fold into its native conformation in the presence of Spy.

**Global fitting of the SH3-Spy kinetic data**. To further demonstrate that SH3 can fold while continuously bound to Spy and to determine what the microscopic rate constants of the various folding transitions are in the presence of Spy, we attempted to simultaneously fit the raw fluorescence traces for SH3 in the absence and presence of Spy to different kinetic models. Global fitting of raw kinetic traces in this manner is a rigorous test of a kinetic model because it requires that a single mechanism simultaneously explain both the changes in $k_{obs}$ and kinetic amplitude as a function of Spy/urea concentration for all of the experiments. With global fitting, one must avoid overfitting the data to overly complex models. In our case, the models that we tested are contingent upon the assumption that SH3 folding follows a two-state folding mechanism, which SH3 has previously

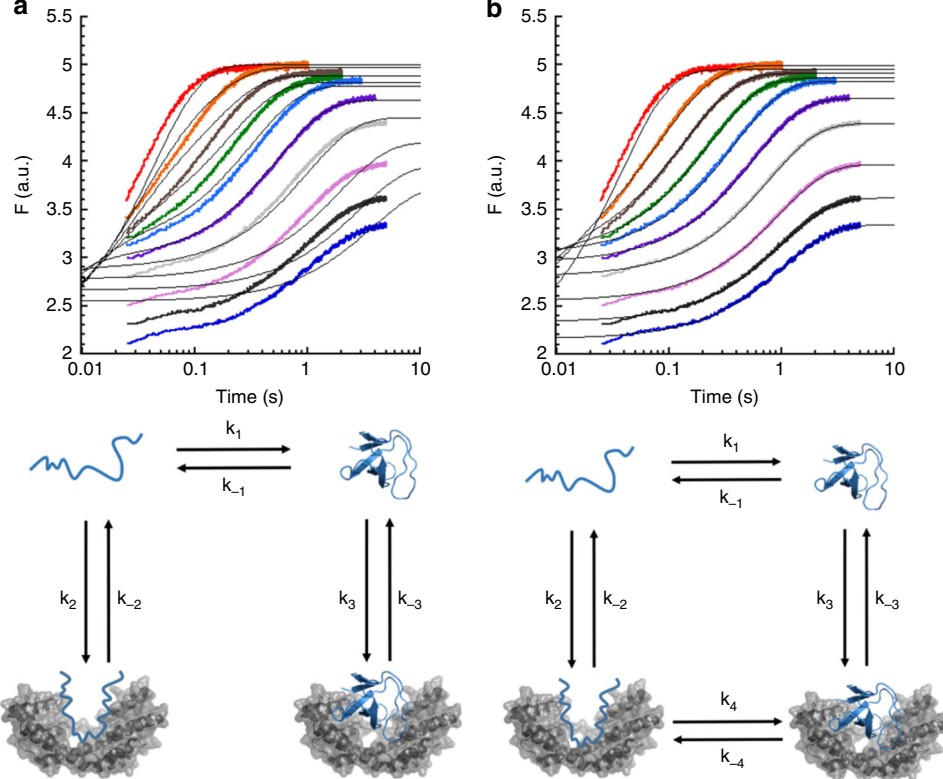

**Fig. 3** Global fitting of the SH3-Spy folding kinetic data. Fitting all the kinetic traces to **a** the mechanism that omits protein folding while bound and **b** the mechanism that allows protein folding while bound. The kinetic models for SH3 folding in the presence of Spy were built upon SH3's two-state folding mechanism. The folding ($k_1$) and unfolding ($k_{-1}$) rate constants of SH3 in the absence of Spy were determined using urea-dependent folding and unfolding kinetics (Supplementary Fig. 7a, b). The rest of the steps in the mechanism were determined by the kinetic experiments for SH3 refolding in the presence of Spy and SH3-Spy binding. For clarity, only the traces for SH3 refolding in the presence of Spy are shown. Complete global fitting results are shown in Supplementary Fig. 8a. The black lines are the best fit to each set of fluorescence traces. SH3 is shown in blue, and Spy is shown in gray in the proposed mechanisms

### Table 1 Rate constants derived from the global fitting

|  | SpyWT | SpyH96L | SpyQ100L |
|---|---|---|---|
| $k_1$ (s$^{-1}$) | 26.7 ± 0.1 | 25.6 ± 0.1 | 27.5 ± 0.1 |
| $k_{-1}$ (s$^{-1}$) | 0.0290 ± 0.0001 | 0.0298 ± 0.0001 | 0.0236 ± 0.0002 |
| $k_2$ (µM$^{-1}$ s$^{-1}$) | 14 ± 1 | 23 ± 1 | 171 ± 8 |
| $k_{-2}$ (s$^{-1}$) | 39 ± 3 | 17.4 ± 0.7 | 6.8 ± 0.3 |
| $k_3$ (µM$^{-1}$ s$^{-1}$) | 12 ± 8 | 91 ± 10 | 60 ± 2 |
| $k_{-3}$ (s$^{-1}$) | 1000$^a$ | 1000$^a$ | 1000$^a$ |
| $k_4$ (s$^{-1}$) | 0.80 ± 0.01 | 0.224 ± 0.001 | 0.0190 ± 0.0003 |
| $k_{-4}$ (s$^{-1}$) | 0.03 ± 0.01 | 0.004 ± 0.001 | 0.0068 ± 0.0004 |
| $K_{d \text{ (unfolded)}}$ (µM) | 2.9 ± 0.3 | 0.76 ± 0.05 | 0.040 ± 0.003 |
| $K_{d \text{ (native)}}$ (µM) | 83 ± 55 | 11 ± 1 | 17 ± 1 |
| $K_{d \text{ (native)}}^{ITC}$ (µM) | 50 ± 6 | 71 ± 19 | 107 ± 25 |

Data were collected under conditions containing 0.83 M urea. The complete fitting results for each data set can be found in Supplementary Fig. 8. $^a$ $k_{-3}$ was fixed at 1000 s$^{-1}$. $K_{d \text{(unfolded)}}$ = $k_{-2}/k_2$; $K_{d \text{(native)}}$ = $k_{-3}/k_3$; $K_{d \text{(native)}}^{ITC}$ was determined by ITC

been shown to follow free in solution[14,15]; we then added complexity to the model until we identified the simplest kinetic mechanism that was consistent with our kinetic data. We first wanted to verify that SH3 follows a two-state mechanism under our conditions, in part because if it does, that would enable a set of experiments that are difficult to do with Im7 because of Im7's more complex three-state folding mechanism. We therefore generated chevron plots for the urea-dependent SH3 folding and unfolding experiments. These experiments directly demonstrate that SH3 does indeed follow a two-state folding mechanism

(Supplementary Fig. 7). We thus were in a position to try fitting our raw kinetic traces to two kinetic mechanisms: one that allows SH3 folding while bound to the chaperone and one that omits the folding while bound step. The mechanism that omits the step for SH3 folding while bound generates a poor fit and thus cannot accurately describe our kinetic measurements (Fig. 3a). In contrast, an excellent fit was achieved for the model that allows folding and unfolding of SH3 to occur while bound to Spy, making this the simplest kinetic model that is consistent with our kinetic data (Fig. 3b and Table 1). Notably, the $K_d$ for Spy binding to native SH3 derived from this global fitting analysis (~80 µM) is similar to the $K_d$ obtained from the ITC experiment (50 ± 6 µM) under identical conditions (Table 1 and Supplementary Fig. 1a). In addition, the sum of the folding- and unfolding-while bound rate constants from the global fitting (~0.83 s$^{-1}$) is very similar to the 0.82 ± 0.01 s$^{-1}$ value of $k_{obs}$ obtained at very high concentrations of Spy (Fig. 1b). These observations strengthen our confidence that the model where SH3 can fold while bound to Spy is operative in our experiments. Notably, the rate constant for SH3 folding while bound to Spy is roughly 50-fold lower than the rate constant for SH3 folding in solution. This reduction in the folding rate is similar to the ~30-fold decrease that we previously observed with Spy-mediated folding of Im7[6].

**Tighter binding to unfolded clients inhibits their refolding.** Similar to what we observed in our Spy–Im7 studies[6], the $K_d$ for Spy binding unfolded SH3 (2.9 ± 0.3 µM, Table 1) is relatively weak compared to most ATP-independent chaperones, which

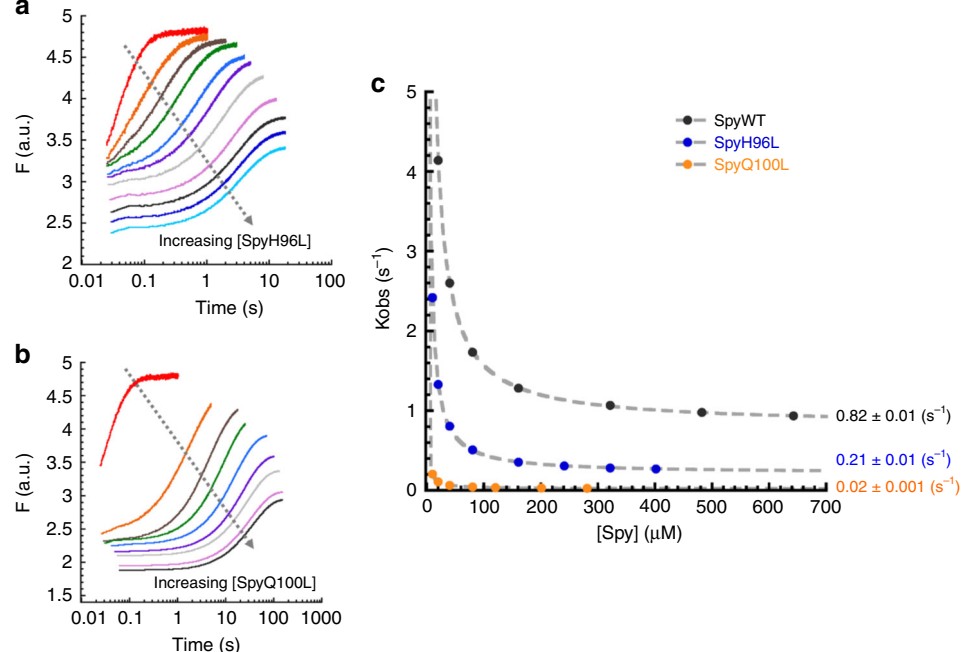

**Fig. 4** The kinetics of SH3 folding in the presence of SpyH96L and SpyQ100L. **a** The fluorescence traces for SH3 refolding in the presence of various concentrations of SpyH96L (0–402 μM dimer after mixing). **b** The fluorescence traces for SH3 refolding in the presence of various concentrations of SpyQ100L (0–281 μM dimer after mixing). All the traces in **a** and **b** can be described by a single exponential function. **c** Plot of $k_{obs}$ for SH3 folding as a function of the concentrations of Spy variants (H96L in blue and Q100L in orange). The $k_{obs}$ hyperbolically decreased with increasing concentrations of the Spy variants, reaching a limiting value that is significantly lower than that of wild-type Spy (black). The saturating value of $k_{obs}$ was 0.21 s⁻¹ for SpyH96L and 0.02 s⁻¹ for SpyQ100L, whereas the value for wild-type Spy was 0.82 s⁻¹. Experimental conditions were exactly the same as in Fig. 2

normally bind clients with nanomolar affinities[5,24–27]. We wondered if weak client binding might be a prerequisite for allowing clients to fold while bound to a chaperone. We therefore decided to examine the refolding kinetics of SH3 in the presence of two Spy variants: SpyQ100L and SpyH96L. These two substitutions, which are located on Spy's client binding site, independently lead to a substantial increase in Spy's binding affinity for unfolded proteins and enhance Spy's anti-aggregation activities[9,10,28]. Analysis of our kinetic data (Fig. 4) shows that the $k_{obs}$ for SH3 folding in the presence of SpyQ100L or SpyH96L reaches saturation at substantially lower Spy concentrations than does wild-type Spy, suggesting that these Spy variants indeed have a stronger binding affinity for unfolded SH3 than does wild-type Spy. We also noticed that the $k_{obs}$ value for SH3 folding using saturating concentrations of SpyQ100L and SpyH96L are 40-fold and 4-fold lower, respectively, than that observed with wild-type Spy (Fig. 4c). This indicates that the rate constant for the folding of SH3 while bound to SpyQ100L or SpyH96L is lower than the value while bound to wild-type Spy.

Similar to the result with wild-type Spy, the best fit was achieved when we globally fitted the kinetic data for SpyQ100L and SpyH96L using the mechanism that allows for SH3 folding and unfolding while bound to the chaperone (Supplementary Fig. 8). Again, the sum of the folding- and unfolding-while-bound rate constants derived from the global fitting analysis (~0.228 s⁻¹ for SpyH96L and ~0.026 s⁻¹ for SpyQ100L) (Table 1) were similar to the $k_{obs}$ values measured at saturating concentrations of SpyH96L (0.21 ± 0.01 s⁻¹) and SpyQ100L (0.020 ± 0.001 s⁻¹) (Fig. 4c). Notably, the global fitting results indicated that the binding affinities for native SH3 are only marginally affected by the Q100L and H96L substitutions, whereas the binding affinity for unfolded SH3 is significantly strengthened by these two mutations (Table 1). Interestingly, we observed an inverse correlation between the $K_d$ for unfolded SH3 ($K_{d\ unfolded}$) and

the rate constants for SH3 folding while bound to the Spy variants ($k_4$); i.e., stronger binding correlates with slower folding. This observation strongly suggests that the ability for clients to fold while bound to Spy mainly relies on a relatively weak chaperone–client interaction.

**SpyQ100L unfolds its native client protein**. Since the Spy variants, especially SpyQ100L, bind more tightly to unfolded SH3 than wild-type Spy does (Table 1), we next examined whether these variants, upon binding, shift the population of SH3 to a less folded conformation. Our simulations suggest that ~30% of SH3 molecules should be shifted to the unfolded conformation in the presence of SpyQ100L, while only ~3% of SH3 should be in the presence of wild-type Spy (Supplementary Fig. 9). To test how closely our simulations match with experiment, we mixed native SH3 with SpyQ100L or SpyH96L in a stopped-flow fluorometer and monitored the change in fluorescence that occurs over time. Because the unfolded state of SH3 is less fluorescent than the native state of SH3, the fluorescence should decrease if SH3 unfolds upon binding to the Spy variants. In contrast to wild-type Spy binding to SH3, which only shows a burst phase (Supplementary Fig. 2), the binding of SpyQ100L or SpyH96L to SH3 proceeds with an additional monophasic decrease in fluorescence after the burst phase (Fig. 5a and Supplementary Fig. 10a). This observation is consistent with an increase in the population of unfolded SH3 upon binding of the Spy variants. In addition, the $k_{obs}$ values for the slow phase in the binding experiments with SpyQ100L and SpyH96L were almost identical to the $k_{obs}$ values obtained in the refolding experiments at the equivalent Spy concentrations (Fig. 5b and Supplementary Fig. 10b), in agreement with the slow phase observed in the binding experiment representing the unfolding of native SH3 after the binding of SpyQ100L.

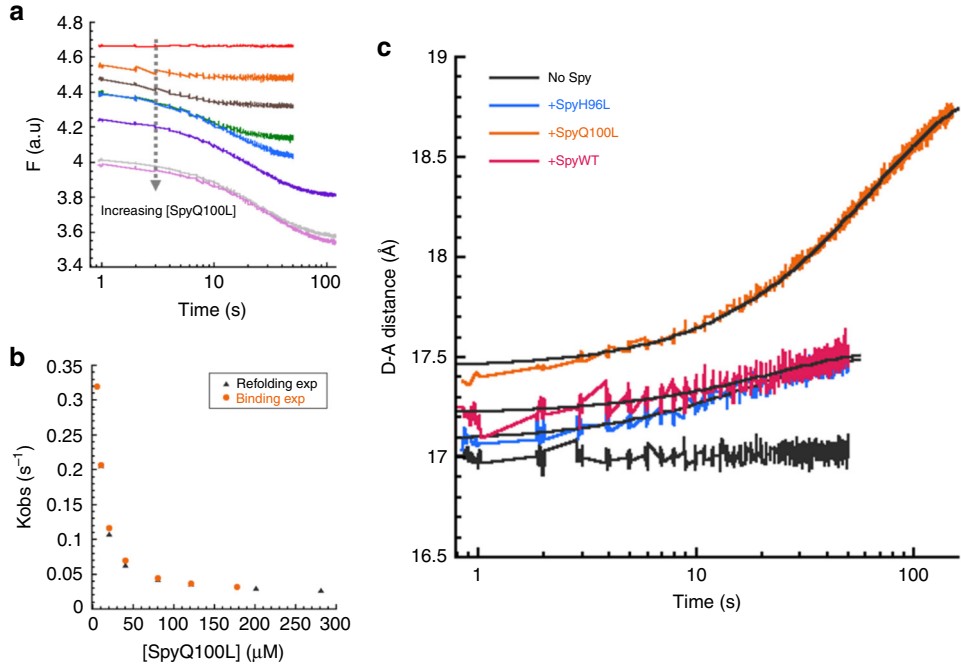

**Fig. 5** The kinetics of SpyQ100L binding to native SH3. **a** Traces for SpyQ100L binding to native SH3. The fluorescence change in the burst phase reflects the rapid binding of SpyQ100L to native SH3. An additional slow exponential decrease was observed after the initial binding of SpyQ100L to SH3, which is not observed with wild-type Spy binding to SH3 (Supplementary Fig. 2). **b** The $k_{obs}$ measured in the binding experiment (orange circles) was overlaid with the $k_{obs}$ determined from the SH3 refolding kinetics (black triangles) as a function of SpyQ100L concentrations. **c** The changes in the apparent D-A distance for native SH3 binding to Spy (160.7 µM after mixing; pink), SpyH96L (160.7 µM after mixing; blue) and SpyQ100L (80.3 µM after mixing; orange). Native SH3 mixed with buffer shown in black. The final apparent D-A distances for SH3 binding to Spy, SpyH96L and SpyQ100L were 17.54 ± 0.03 Å, 17.55 ± 0.03 Å and 18.87 ± 0.02 Å, respectively

To complement the fluorescence binding kinetics for SpyQ100L described above, we monitored the changes in apparent donor–acceptor distance that occur when native TNB-labeled SH3 is mixed with SpyQ100L. We observed an increase in the apparent donor–acceptor distance after SpyQ100L bound to native SH3 from 17.45 ± 0.02 Å to 18.87 ± 0.02 Å (Fig. 5c), which occurred over a similar timescale as the tryptophan fluorescence data in Fig. 5a. In contrast, wild type Spy and SpyH96L binding have minor effects on the structure of native SH3 in terms of the changes in the apparent donor–acceptor distance (Fig. 5c). Our FRET kinetic data thus support the notion that SpyQ100L shift the population of SH3 away from its native state due to their strong binding affinity for the non-native client.

**SpyQ100L negatively impacts bacterial fitness.** We next wondered how the changes in the ability of SpyQ100L to affect the folding of the model protein SH3 are reflected *in vivo*. In particular, we reasoned that if SpyQ100L is able to unfold not just the model protein SH3 but also native proteins present in *E. coli*, the overexpression of SpyQ100L might well have a deleterious effect on *Escherichia coli* growth and/or survival. Notably, the variant H96L is similar to substitutions that have occurred during evolution, as methionine and other hydrophobic residues are found at position 96; however, substitutions similar to the Q100L substitution were never found in Spy orthologues in our search of the UniRef90 database as of March 2019 (Supplementary Fig. 11). This observation indicates that there is a strong selective disadvantage to having leucine or other hydrophobic residues at position 100 even though, *in vitro*, SpyQ100L displays a more potent binding affinity for its clients than any of the other mutants that we selected for their improved ability to stabilize Im7 *in vivo*[28]. To test this hypothesis, we overexpressed Spy and

the H96L or Q100L variants in the periplasm of an *E. coli* Δ*spy* strain (SQ1731) and measured *E. coli* growth over time. Although the growth rate during log phase was not significantly different between the various strains, the final O.D.$_{600}$ reached by the Q100L strain in the late stationary phase was significantly lower than that reached by the other Spy overexpressors, indicating that the Q100L expression is particularly disadvantageous in stationary phase compared to wild-type Spy (Fig. 6a).

Periplasmic proteins are usually extremely stable[29,30], and Spy is only overexpressed in response to unfolding stress[31]. We reasoned that since cytosolic proteins are generally less stable than periplasmic ones, Spy expression in the cytosol would give it the potential to bind many more clients and therefore might produce more striking phenotypes than this stationary phase phenotype. Indeed, we found that cytosolic Spy expression negatively impacts both the log phase growth rate and the final O.D. reached in stationary phase (Fig. 6b). The most significant effect is on the final O.D.$_{600}$ with SpyQ100L expression. Microscopic observation showed that the *E. coli* strain expressing cytosolic SpyQ100L has an extremely filamentous morphology (Fig. 6c), which may also account for the poor growth rate and unusual clumping of these cultures (Supplementary Fig. 12). The strains expressing wild-type Spy or SpyH96L also showed elongated cell lengths, but this morphology was much severe as compared to the strain expressing SpyQ100L. The strain harboring the empty vector and all of the strains under non-induced conditions were normal in size and morphology. In summary, our *in vivo* data suggest that the cytosolic expression of Spy and its variants, especially Q100L, can severely affect cellular morphology. One possibility is that these Spy mutants may be able to unfold Fts (filamentous temperature-sensitive) cytoplasmic components that are involved in cell division.

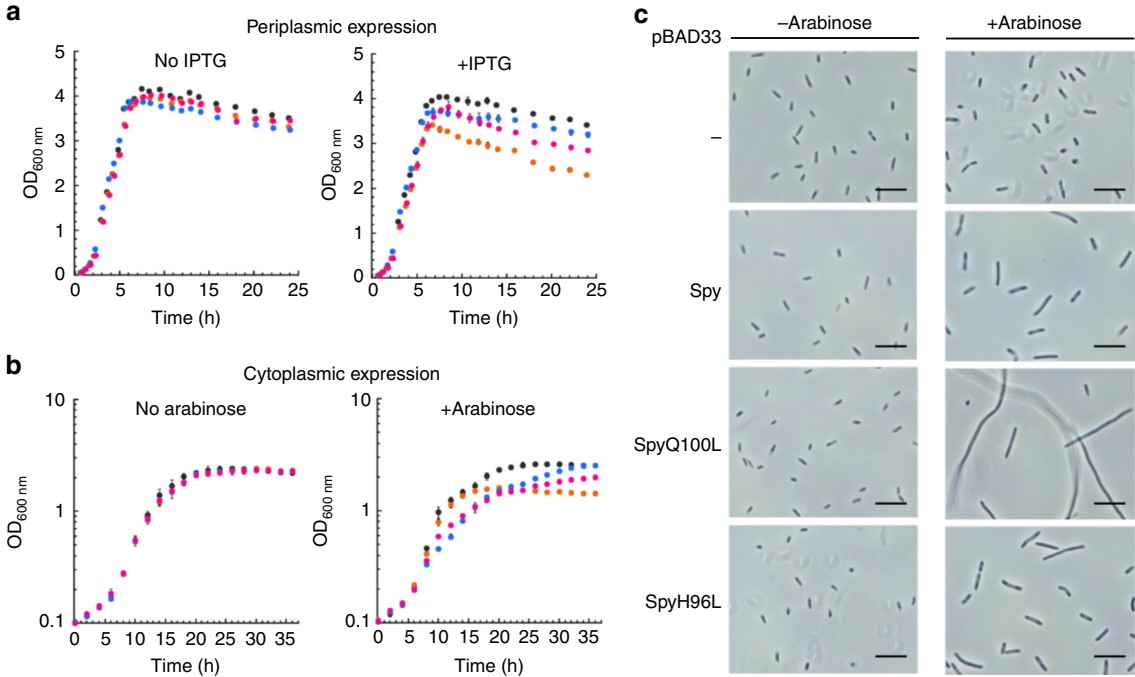

**Fig. 6** Physiological effect of overexpression of Spy and its variants (Q100L and H96L) on *E. coli* growth curve. Spy (pink) and its variants (SpyH96L: blue; SpyQ100L: orange) were overexpressed in the periplasm of *E. coli* **(a)** and in the cytoplasm of *E. coli* **(b)**. The black arrow in the curves on the right indicates the time point of adding inducer (IPTG or arabinose). **c** Cell morphology of *E. coli* expressing Spy and its variants (scale bar 10 μm)

## Discussion

ATP-independent chaperones have generally been thought to tightly hold onto unfolded clients in order to prevent their aggregation, but to be incapable of facilitating their refolding[4]. Using Spy as a simple model system for chaperone action, we have obtained evidence for another chaperone mechanism—one that allows proteins to fold while they remain loosely but continuously in contact with the surface of the chaperone. Previously, we reported that the all α-helical protein Im7 can fold while continuously bound to Spy[6]. In this work, we show that another folding model protein, Fyn SH3, whose native structure consists of two antiparallel ß-sheets, can also fold while bound to Spy. This study implies that allowing clients to fold while bound may be a general mechanism for Spy-mediated protein folding. Our FRET analysis indicates that binding of Spy to unfolded SH3 induces structural compaction. Similar observations have been reported in Spy–Im7 complex using NMR[7,8].

Previously, we used a genetic approach to identify a set of Spy mutants by their enhanced ability to stabilize Im7 *in vivo*[28]. Not surprisingly, most of these variants were found to bind more tightly to Im7 *in vitro* and showed an enhanced ability to inhibit protein aggregation of model substrates. While most of these mutants have been found in evolution, leucine was never observed at position 100 in our PSI-BLAST search of Spy sequences. This finding strongly indicates that Q100L is highly disfavored in evolution. Here, we used SpyQ100L and an evolution-favored mutant, H96L, to ask if there is any evolutionary disadvantage for Spy to have a strong binding affinity for its clients. Using SH3, which has a simple two-state folding mechanism, we were able to show that there is an inverse relationship between the strength of Spy's client binding affinity and its ability to allow proteins to fold while bound. The tighter the interaction is between the chaperone and the unfolded client, the more the chaperone slows the folding rate of its bound client. Due to its increased binding affinity for the unfolded SH3, we also observed that SpyQ100L displays a much stronger tendency to

unfold native SH3 than other mutants do. In agreement with our in vitro evidence, expression of SpyQ100L negatively affects both bacterial growth and morphology, suggesting that Spy's client binding affinity must be limited to the stage where it becomes too strong to release its clients effectively and to allow them to fold effectively while bound. Both of these chaperone-mediated actions are likely to be crucial during recovery from stress.

A folding-while-bound mechanism is also used by other chaperones[11]. One of these is SecB, a chaperone which maintains nascent polypeptides destined for the cellular envelope in an unfolded state and transports them to the inner membrane for translocation. While SecB blocks the folding of its client, maltose binding protein containing its signal sequence (pre-MBP)[32,33], SecB does not completely inhibit the folding of mature MBP, which lacks the signal sequence. The reason why MBP and pre-MBP have different folding behaviors in the presence of SecB is still being debated[27,33]. One plausible explanation is that MBP can fold while bound to SecB whereas pre-MBP cannot[33]. Based on the inverse relationship between chaperone binding affinity for the clients and the ability to allow protein folding while bound, we thought that the strength of SecB-client interactions might significantly affect the folding rate constant for MBP while it is bound to SecB. This idea is supported by the fact that the binding affinity of SecB for pre-MBP is 100-fold stronger than that for mature MBP[34]. Further kinetic experiments are required before this hypothesis can be verified.

Another ATP-independent chaperone that could potentially use this mechanism to mediate protein folding is trigger factor[35]. The crystal structure of the trigger factor-substrate complex revealed that trigger factor, like Spy, has a highly hydrophilic client-binding surface, albeit with an opposite charge. The trigger factor-bound substrate displays a near-native structure, indicating that trigger factor not only binds to nascent peptides but also folded proteins. In addition, the UV-crosslinking experiments showed that trigger factor could accommodate unfolded and folded state of small protein[36]. These lead to the possibility that at

least some proteins can fold while bound on the surface of trigger factor. Instead of just being responsible for the "holding" of unfolded or misfolded proteins, our observations and reports in the literature are consistent with the notion that ATP-independent chaperones could provide their clients with a folding friendly surface, one that loosely binds proteins and in doing so provides them with the opportunity to search for their native conformations.

This chaperone mechanism that assists protein folding might not be exclusively utilized by ATP-independent chaperones. For example, the chaperonin GroEL and Hsp70 have also been reported to allow their clients to undergo conformational sampling while chaperone bound[37–40]. Folding-while-bound may represent an ancestral mechanism of chaperone-mediated folding that requires a relatively weak chaperone–client interaction. We postulate that this primitive mechanism may conceivably be lost in evolution as the result of an increase in the affinity of the chaperone for its clients in at least two ways. One way could be through the evolutionary acquisition of cofactors or co-chaperones capable of regulating client binding and release; for instance, in the GroEL/GroES/ATP system[41]. Another way folding-while-bound may be lost would be to substantially increase the affinity of chaperones for non-native clients. This would effectively convert these folding chaperones to "holding" chaperones, where client release does not occur unless mediated by other "folding" chaperones or by stress relief-mediated changes in chaperone function, as is seen for redox-regulated chaperones (e.g., Hsp33)[42,43], for acid-responsive chaperones (e.g., HdeA and HdeB)[44–47] and for heat-induced chaperones (e.g. sHSPs)[4,48]. Notably, sHSPs have been considered to be an ancestor of most chaperones[48]. In contrast to the chaperone mechanism that we proposed here, most sHSPs are able to regulate the binding and releasing of their clients through the structural oligomerization[49–51]. We thus surmise that the folding-while-bound mechanism may be an alternatively prehistoric method for chaperoning protein folding before evolution developed the advanced regulatory mechanisms that make chaperones more efficient aggregation inhibitors and folding catalysts.

## Methods

**Protein expression and purification.** The plasmids with gene encoding either wild-type SH3 and the variants or wild-type Spy and Spy variants (SpyQ100L and SpyH96L) hereafter in this method referred to as protein were expressed and purified as follows. The protein encoding plasmids were transformed individually into *E. coli* BL21 (DE3) cells (New England Biolabs) and expressed by addition of IPTG to 0.1 mM when the strain had reached an O.D.$_{600}$ of 1.0. After 16 h of continued incubation at 20 °C, cells were pelleted and resuspended in lysis buffer containing 50 mM sodium phosphate (pH8.0), 400 mM NaCl, 10% glycerol and 0.05 μg/ml DNase. Cells were then lysed using a French press for 2 cycles at 1300 psi. Cell debris was pelleted at 36,000 g for 30 min at 4 °C twice, and the supernatant from the second centrifugation was loaded onto a Ni-His Trap column (GE Healthcare) with 20 mM imidazole and washed with 50 ml of lysis buffer containing 30 mM imidazole. Histidine tagged proteins were then eluted using 30 ml of lysis buffer containing 500 mM imidazole. The His-SUMO tag attached to the N terminus of the proteins were removed by addition of 500 μg of ULP1 protease. The proteins were then dialyzed against 0.5 liters of 40 mM Tris (pH 8.0) and 400 mM NaCl at 4 °C overnight. After dialysis, the sample was passed through a 5 ml Ni-His Trap column equilibrated with dialysis buffer containing 30 mM imidazole to bind and remove the cleaved His tag. The flow through was diluted 5-fold using buffer containing 25 mM Tris, pH 8.0 and was then loaded onto a HiTrap Q column (GE Healthcare). The proteins were eluted with a NaCl gradient (83.5–415 mM) in 25 mM Tris (pH 8.0). After concentrating them, the fractions containing the purified protein of interest were then purified with a HiLoad Superdex 75 column (GE Healthcare) equilibrated with 40 mM HEPES (pH 7.5) and 50 mM NaCl (HN buffer). The purified sample of protein was frozen in liquid nitrogen and stored at −80 °C. Plasmids with gene encoding wild-type Spy and Spy variants (SpyQ100L and SpyH96L) were from a lab collection[6]. The gene for wild-type *Gallus gallus* Fyn SH3 domain (T85-D142) was a kind gift of Lewis Kay. 7 residues (GAMVQIS) and a single arginine were added to N terminus and C terminus, respectively[52], using primer 1 (5′ CT CAC GGA TCC GGA GCT ATG GTA CAA ATT TCT ACT CTT TTT GTG GCG CTT TAT GAC 3′) and primer 2 (5′ CA

TAG CTC GAG TTA GCG ATC GAC TGG AGC CAC ATA ATT ACT G 3′) and ligated into pET28b-based vector with an N-terminal His$_6$-SUMO tag[12]. Fyn SH3 T2C mutant was generated by site-directed mutagenesis using primer 3 (5′ ATC CGG AGC TAT GGT ACA AAT TTC TTG TCT TTT TGT GGC GC 3′) and primer 4 (5′ GCG CCA CAA AAA GAC AAG AAA TTT GTA CCA TAG CTC CGG AT 3′).

**Preparation of TNB-labeled SH3.** The TNB-labeled SH3 was obtained by incubating the purified protein with a 100-fold excess of 5′, 5′-dithiobis (2-nitrobenzoic acid) (DTNB) in HN buffer at room temperature for 45 min The excess dye was separated from the labeled proteins by using PD−10 column (GE Healthcare), followed by the HiLoad Superdex 30 pg prepacked column (GE Healthcare). The concentration of labeled protein was determined as follows, exactly as done previously[53]. Since the TNB adduct contributes to absorbance at 280 nm, absorbance at this wavelength could not be used directly to determine concentrations of labeled proteins. To first determine the extent of this contribution to absorbance at 280 nm, a solution of labeled protein was divided into two parts. One of these parts was treated with DTT, while the other part was treated with an equal volume of buffer. Each part was then desalted using an AKTA Chromatography system, and the absorbance at 280 nm of each solution was determined. It was found that TNB adduct labeled in Cys 2 increased the absorbance at 280 nm by 18.9%. Thus, this effect needs to be considered to calculate the accurate concentration of TNB-labeled SH3.

**Isothermal titration calorimetry.** The thermodynamic analysis of Spy-SH3 interaction was performed using a MicroCal iTC200 (Malvern Instruments) at 25 °C. The buffers of all samples were dialyzed against HN buffer containing 0.83 M urea overnight. 2.4 mM SH3 in the syringe injector was titrated into a calorimeter cell containing 100 μM wild-type Spy (or Spy variants). For each titration, the injection volume was 1 μl and the injection interval was 300 s. The heat of dilution for SH3 was removed by subtracting data for a blank titration of SH3 into buffer. The ITC thermograms were fitted to a one-site model using the MicroCal Origin software.

**Stopped-flow kinetic experiments.** All kinetic measurements were monitored in a KinTek SF-300X stopped-flow instrument at 25 °C. For Tryptophan fluorescence, two tryptophans (Trp36 and Trp37) of Fyn SH3 were excited at 296 nm and the emitted fluorescence was monitored using a 320 nm long-pass filter. SH3 refolding in the absence of Spy was initiated by making a11.5-fold dilution of SH3 that was in HN buffer containing 9.5 M urea into the same buffer containing various lower concentrations of urea. The final concentration of SH3 in each case was 4.8 μM, unless specifically mentioned otherwise. Each kinetic trace was acquired as an average of 15–20 individual traces. Two main kinetic phases were observed in the refolding experiment: a fast phase, completed within 5 s, accounting for over 97% of total fluorescence change; and a slow phase, which displayed a urea-independent rate constant ($\sim$0.3 s$^{-1}$) caused by proline isomerization[15]. Due to its small amplitude, the slow phase corresponding to the proline isomerization was not considered in our analysis. Unlike the refolding of wild-type SH3, the refolding of SH3T2C shows a significant proline isomerization step ($\sim$11% of the total change). Although this implies that threonine-to-cysteine substitution affects the cis-trans proline isomerization, the refolding rate constant measured in the major phase of refolding of both SH3 variants is similar. The urea-dependent SH3 unfolding experiments were conducted in the same way, except the initial protein solution was prepared in the urea-free HN buffer. All the fluorescence traces in the unfolding experiments were well described by a single exponential function. For the experiments measuring the folding of SH3 in the presence of Spy variants, a 1:10.5 mix was used to mix 4.8 μM SH3 in HN buffer containing 9.5 M urea with HN buffer containing various concentrations of wild-type Spy (0–643 μM dimer after mixing), SpyH96L (0–402 μM dimer after mixing) and SpyQ100L (0–281 μM dimer after mixing). For the experiments measuring the binding of native SH3 to Spy, a 1:10.5 mix was used to mix 4.8 μM SH3 in HN buffer containing 0.83 M urea with various concentrations of wild-type Spy (0–229 μM dimer after mixing), SpyH96L (0–321 μM dimer after mixing) and SpyQ100L (0–177 μM dimer after mixing). The urea was added in the HN buffer so that the buffer conditions were identical to the one used in the experiments of SH3 folding in the presence of Spy. The traces for the SH3-Spy binding were monitored in the stopped-flow with the auto-shutter on to reduce the effect of photobleaching during the course of experiment. Six to ten individual traces were averaged for the displayed fluorescence measurements in the plot. The background fluorescence of Spy was subtracted from each kinetic trace at each Spy concentration. All the kinetic traces were fitted with a single exponential function to obtain the observed rate constant using KaleidaGraph (Synergy Software).

For the stopped-flow FRET experiments, the refolding reactions of unlabeled SH3 and TNB-labeled SH3 and the binding experiment of SH3 to Spy were carried out in a similar manner as the measurement of tryptophan fluorescence, except the emission was monitored using 340 ± 10 nm band-pass filter. The experiments were repeated in triplicate and the estimated uncertainty is given as the apparent deviation (N = 3). Each kinetic trace was acquired as an average of 3–4 consecutive traces. The FRET efficiency (E) and apparent distance (R) at each time point during

refolding were calculated using Eq. (1):

$$E = 1 - \frac{F_{DA}}{F_D} = \left(1 + \frac{R^6}{R_0^6}\right)^{-1} \quad (1)$$

where $F_D$ is the fluorescence intensity of unlabeled SH3, $F_{DA}$ is the fluorescence intensity of TNB-labeled SH3 and $R_0$ is Förster distance, which was determined experimentally as described below. The kinetic traces for SH3 refolding can be described by two-exponentials, while the ones for SH3 binding to Spy can be fit by single-exponential. The values of the initial and final apparent D-A distance mentioned in main text and figures were obtained by the fitting results.

The value of $R_0$ can be determined by using Eq. (2):

$$R_0 = 0.211[Q_D J \kappa^2 n^{-4}]^{\frac{1}{6}} \quad (2)$$

where $Q_D$ is the quantum yield of donor fluorescence in the absence of acceptor, J is the overlap integral of donor emission and acceptor absorbance, $\kappa^2$ is the dipole orientation factor, and n is refractive index of the solvent. The values of all parameters in Eq. 2 are shown in Supplementary Table 2. The values of $Q_D$ in different conditions were determined according to the article "A guide to recording fluorescence quantum yields", using N-acetyltryptophanamide in water (Q = 0.14) as standard solution. The resulting quantum yields for native SH3 in 0 M urea and unfolded SH3 in 9.5 M urea are 0.18 and 0.13, respectively. The value of J was calculated in the UV-Vis-IR Spectral software, a|e. The value of 2/3 for $\kappa^2$ was used in the calculation. The values of n for the native condition and unfolding condition are 1.33 and 1.41, respectively[54]. The values of $R_0$ for native SH3 and unfolded SH3 are 25.0 Å and 22.9 Å, respectively, which are similar to the values that determined previously[18–20]. A mean value of 24.0 Å for $R_0$ was used for all the calculations in this study.

**Global fitting of kinetic data**. The global fitting analysis was conducted in KinTek Explorer (Version 6.2.170308) and followed the approach described previously[6] with minor revisions due to the different folding mechanism of SH3 (two-state) versus Im7 (three-state). Briefly, the urea-dependent folding and unfolding experiments were fitted on the basis of the assumption that the folding/unfolding rate constant ($k_f$ and $k_{uf}$) varies with the concentration of urea according to the Eqs. (3) and (4):

$$k_f = k_f^{H2O} e^{(mf/RT)[urea]} \quad (3)$$

$$k_{uf} = k_{uf}^{H2O} e^{(muf/RT)[urea]} \quad (4)$$

where $k_f^{H2O}$ and $k_{uf}^{H2O}$ is the folding/unfolding rate constant measured in 0 M urea, mf and muf defines the urea-dependent change in the folding/unfolding rate constant, R is the gas constant, and T is temperature in Kelvin. The fluorescence of unfolded SH3 ($SH3^U$) and native SH3 ($SH3^N$) can be described by 0.404 + 0.016* [urea] and 0.985 + 0.027*[urea], based on the assumption that fluorescence changes linearly with the concentration of urea. The initial estimation for mf, muf, $k_f^{H2O}$ and $k_{uf}^{H2O}$ were obtained from the chevron plot for SH3 folding and unfolding (Supplementary Fig. 7c). Once the best fit for the SH3 folding and unfolding kinetics was achieved, the kinetic experiments for the folding of SH3 in the presence Spy and binding of SH3 to Spy were included in the fitting to test the different mechanisms. Since the intensity of fluorescence traces for SH3 folding in the presence of Spy are severely reduced, especially at high concentrations of Spy due to the inner filter effect, significant scaling factors were required to correct for the intensity of the traces with high concentrations of Spy dimer (above 100 μM). To help constrain the fitting process, we fixed the dissociation rate constant ($k_{off}$) for the Spy-native SH3 binding step to 1000 s$^{-1}$ since the step for Spy binding to native SH3 is too fast to be observed in the stopped-flow; this allows the global fitting algorithm to simply fit for the binding affinity of this step. The best fitting results that could satisfactorily describe the data were evaluated by visual inspection of the fit.

**Growth curve analysis**. To express Spy in the periplasm, the nucleotide sequence of Spy with the signal sequence was cloned into the pCDFTrc vector. Q100L and H96L missense mutations in Spy were generated by site-directed mutagenesis[28]. The constructs were transformed into *E. coli* MG1655 Δ*hsdR* Δ*spy* K-12 strain. Overnight cultured strains were reinoculated into 20 ml of LB medium in a 250 ml flask with 100 μg/ml chloramphenicol as the starting O.D.$_{600}$ 0.04. Cells were grown at 37 °C with shaking at 200 rpm. The overexpression of Spy proteins was initiated by adding 1 mM of IPTG at mid-log phase (O.D. ~0.3). O.D.$_{600}$ were measured every 30 min in triplicate (N = 3) until stationary phase and after that O. D.$_{600}$ were measured every 2 h.

To express Spy in the cytoplasm, the nucleotide sequence of Spy without the signal sequence (amino acids 24–161) was cloned into the pBAD33 vector. Q100L and H96L missense mutations in Spy were generated by site-directed mutagenesis. The constructs were transformed into *E. coli* MG1655 Δ*hsdR* Δ*spy* K-12 strain. Overnight cultured strains in LB medium were harvested and washed with M9 medium supplemented with 0.01% thiamine and 0.4% glycerol. 1% L-arabinose was added to induce the expression of Spy proteins. Cells were reinoculated into 20 ml of M9 medium in a 125 ml flask as O.D.$_{600}$ 0.1 and grown at 37 °C with agitation of 200 rpm. O.D.$_{600}$ were measured every 2 h in triplicate (N = 3).

**Microscopy**. For cell imaging, 3 μl of the cell culture at stationary phase (35 h) was added to a glass plate and covered with a cover slide. Differential interference contrast (DIC) images were captured using an ECLIPSE E600 (Nikon) microscope with 100X magnification.

**Sequence alignment**. UniProt Clusters 90% databases were used with PSI-BLAST to identify Spy homologs using *E. coli* Spy as a reference sequence. In total 200 sequences with an E value ≤ 0.001 were aligned with ClustalW using default settings and generated a phylogenetic tree. 134 sequences were selected as the Spy clade, while the remaining belong to the CpxP clade. The selected sequences were then aligned with ClustalW and the results were displayed with WebLogo.

**Reporting Summary**. Further information on research design is available in the Nature Research Reporting Summary linked to this article.

## Data availability
The source data underlying Figs. 1a–b, 2a–b, 3a–b, 4a–c, 5a–c and 6a-band Supplementary Figs. 2, 3a–b, 4, 5b–e, 6a–c, 7a–c, 8a–c and 10a–b are provided as a Source Data file. Other data that support the findings of this study are available from the corresponding author upon reasonable request.

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

## Acknowledgements
We thank Ursula Jakob and Susan Marqusee for helpful discussions and Dan Raleigh for his advice on the FRET experiments and useful comments on the manuscript. We thank Lewis Kay for supplying us with the gene for wild-type *Gallus gallus* Fyn SH3 domain This work was funded by the Howard Hughes Medical Institute.

## Author contributions
K.W., F.S., C.L. and J.C.A.B. Experimental design and data analysis, K.W., F.S. and J.C.A.B. manuscript preparation and study concept

## Competing interests
The authors declare no competing interests.
