## [Peer Review File · Nature Communications]

Reviewers' comments:

Reviewer #1 (Remarks to the Author):

In their paper "protein folding while chaperone bound is dependent on weak interactions" Wu et al., used two mutants of the ATP-independent periplasmic chaperone Spy and they describe a new Spy client (substrate), SH3, to further characterize the mechanism by which the chaperone accelerates its client's refolding to their native state. They measured the kinetics of the client intrinsic fluorescence in the presence of increasing Spy concentrations. The global fitting of the data strongly suggested that, as they previously established for the Im7 substrate, SH3 too is refolding while bound to the chaperone and not after dissociation. They characterized two Spy mutants, H96L and Q100L, that bound the native state of SH3 much tighter than WT Spy, and this reduced the rates of Spy-mediated native refolding. They complemented their study with kinetics of ThT fluorescence of SH3mut Δ 57 fibril formation and showed that the SpyQ100L with the highest affinity for the native state effectively prevented amyloid formation of the SH3 variant. Moreover, excessively tight binding of native SH3 by SpyQ100L seemingly caused a slow unfolding. Overall, the data strongly indicates central the role of weak interactions between Spy and the natively folded terminal intermediate of the folding reaction which is being accelerated by the Spy chaperone. Overexpression showed the biological relevance of the abnormally high-affinity of the Spy Q100L mutant for its substrate(s), which caused severely impaired cell division. This paper provides important new in-depth insights on the molecular mechanism of a molecular chaperone that does not need ATP hydrolysis to accelerate and increase the yields of native refolding of two chemically pre-unfolded clients. Although some aspects of the Spy mechanism have been described previously with the Im7 client, notably the refolding of the client while still bound to the chaperone, SH3 is a very different client, which has a more complicated folding pathway. Hence, this work provides a much-needed additional example to generalize the Spy chaperone mechanism. Moreover, the two Spy mutants with abnormally high affinities for their clients, confirmed a general principle in enzymology that high affinity for the substrate is not always a corollary to high and effective enzymatic activity, especially when high affinity of a bound late intermediate may retard the clearing of the active site, thereby preventing the iterative processing of other substrates.

Major concern:

The evidence that SH3 folding occurs while bound to the chaperone is convincingly, albeit indirectly inferred from the best fits of the kinetic curves. Other experimental approaches should be undertaken to show more directly that the native refolding of SH3 occurs while bound to Spy and not following dissociation. One obvious approach would be to engineer an SH3 with two bound low molecular weight fluorophores allowing assessment by FRET of the different possible states of the client protein: native, unfolded, misfolded and/or aggregated. This would likely confirm by single molecule studies that SH3 folding occurs while bound to Spy. Other methods should be considered, such as chemical crosslinking and mass spectrometry, limited proteolysis-coupled to mass spectrometry, BIFC, SPR, etc.

Minor concern:

The authors should try to discuss more in depth the implications of their results for the chaperone mechanism of action on its clients. For example, on the one hand, the refolding kinetics strongly suggested that the pre-unfolded SH3 was led to rapidly reach its native state while bound to the chaperone. Thus, in principle, if the stoichiometry of the reaction was set to undergo a single turn over, a slower dissociation rate of the natively refolded client from the high-affinity chaperone mutants, should not affect the rate and the yield of Spy-mediated refolding of SH3 to the native state, as observed. If several turnovers of more than a single client per Spy active site, were necessary to produce the observed refolding rates, this could explain the lower values observed with the high affinity Spy mutants. Slower active site clearance would impair iterative processing of the intermediate species. Consequently, the paper would gain from showing kinetics of SH3 refolding to the native state under conditions where the unfolded SH3 would be in a large molar excess over Spy.

Another central question of this system is what is the precise state of the pre-denatured substrate

when, following dilution of the denaturant, it first interacts with the Spy? Is it mostly unfolded? Is it mostly already misfolded (because of a rapid hydrophobic collapse) or would it be aggregated? The finding that SH3mut Delta57 fibril formation was prevented by SpyQ100L strongly suggests that even WT Spy could bind partially structured misfolding and aggregating intermediates, and not only as generally presumed, to unfolded entrant substrates, slowly turning thereafter into late bound natively refolded intermediates.

Examination of Fig 6 suggests that the SpyQ100L mutant binds the native SH3 so strongly that following initial binding, Spy could be causing slow structural changes in the bound protein that could be interpreted as partial unfolding. This would suggest that following initial binding of the denatured client to WT Spy, the latter would exert an unfolding action on the substrate, as shown in several instances to be the case with the ATP-dependent chaperones GroEL, DnaK and Hsp104. Could the Spy-mediated unfolding of misfolded clients explain the observed slow exponential decrease of fluorescence in Fig 6? In contrast, if the entrant substrate was to be predominantly unfolded at the time of binding, the slow exponential decrease of fluorescence could result from chaperone-mediated compaction, since further unfolding of an already unfolded polypeptide is a less sound possibility.

Here again, knowing the precise state of the pre-denatured substrate and its fate during initial binding, and during the subsequent slow exponential phase and until it has finally reached its native state, while bound, could/should be addressed by FRET with double-labelled SH3 and/or Im7.

In the discussion, trigger factor and Hsp33 are mentioned and the authors discuss about a possible ancestral mechanism of chaperone mediated folding without ATP that would require weak chaperone-interactions. This mechanism would have appeared in evolution before the complex GroEL/GroES mechanism, which can use the energy of ATP hydrolysis to unfold and possibly also to drive the dissociation of over-sticky clients. Whereas, this is probably true in most of the cases, very early reports from the Fersht lab showed that some proteins could be folded without ATP hydrolysis, solely by single or seven fused GroEL apical domains (called mini-chaperones). Moreover, small HSPs, which like Spy are not ATPase chaperones, should also be mentioned in this discussion. They have been considered by many to be the most ancestral of all chaperones as they are present in all organisms, plants, fungi, metazoans, eubacteria as well as the most primitive forms of Archea. The last universal common ancestor (LUCA) was predicted to contain only sHSPs as Hsp60 chaperonins to control proteostasis. LUCA was missing Hsp70, Hsp90, Hsp100 and notably also Spy. sHSPs likely work by way of maintaining weak interactions with their clients but this remains to be experimentally established.

Reviewer #2 (Remarks to the Author):

The manuscript by Wu et al. exploits stopped-flow kinetics experiments to investigate the interaction between the ATP-independent periplasmic chaperone Spy and a client SH3 domain. Based upon their data, the authors propose that the stronger the interaction is, the slower is the client's refolding.

In general, I found that the topic is of interest, especially because the idea of a chaperone mediating refolding while bound to its client, rather than after its release, is novel. The results are clear. However, I found that the manuscript lacks the details and the resolution that would be necessary to support authors' conclusions in a high impact journal. Apart from the ITC in the first section, the manuscript takes advantage of but one experimental probe, tryptophan intrinsic fluorescence, which can be monitored only with cut-off filters in stopped flow experiments. Second, I have personally employed kinetic analyses a lot and I know it is a very useful tool, yet it never provides direct evidence, it just tells which is the most likely among a set of models examined. More specific comments:

. I think the authors should describe more clearly their probe. How many tryptophans does SH3

have? Where are these located? I guess Spy does not have any tryptophan or this would affect data interpretation. This should be briefly discussed.

. I have several doubts about the ThT-amyloid section and it is not very clear to me how these results fit into the rest of the manuscript. The authors take advantage of a variant of SH3, known to be aggregation prone. Spy inhibits aggregation and the authors conclude that these results are consistent with Spy providing a folding-friendly environment. However, no one knows how this inhibition is achieved and the variant employed here was reported to be a three-state folder in Neudecker et al. 2012, with a poorly populated partially folded aggregation prone state. Most likely, Spy binds the folded state (something the authors show for Q100L and H96L), thus shifting the equilibrium and inhibiting population of the aggregation-prone intermediate. What one should investigate here is whether, starting from the unfolded state, Spy is able to suppress the SH3 amyloidogenic intermediate. However, the rest of the manuscript investigates a two-state folder and thus it is not very clear to me what information these experiments provide, with respect to the manuscript main message.

. The authors claim that Spy Q100L and Spy H96L unfold their clients based upon the decrease in fluorescence that was observed following binding. How can the authors rule out a structural rearrangement, different from unfolding and involving a decrease in fluorescence? Indeed, tryptophans get commonly exposed to the solvent during unfolding and this involves red-shift and an increase in fluorescence above 320 nm, even though this is not a strict rule. The authors point out that the unfolded state of SH3 is less fluorescent than the native state, but I guess this was observed at high [denaturant]. The authors should extrapolate the fluorescence signal emitted by the U state at the denaturant concentration at which they did the binding to Q100L/H96L and determine whether such signal is compatible with that observed at the end of the slow phase.

. Like the amyloid section, the bacterial fitness section reads a little out of place in this manuscript. If it is intended to suggest that the folding-while-bound capability of Spy enhances fitness, this conclusion is clearly far-fetched, and in fact the authors do not state that. If, instead, it aims to investigate how chaperones affect morphology or other processes, it sounds off topic.

. All the experimental values in the text come without experimental errors.

. Missing dimensions in Kd data reported in table 1.

Reviewer #3 (Remarks to the Author):

This is a very nice work where the authors demonstrate that a recently discovered client of Spy (an ATP independent chaperone) can fold while bound to the chaperone.

Thanks to a straightforward analysis of the kinetic data, the authors clearly demonstrate that, challenging the SH3 domain with increasing concentrations of Spy, leads to an observed rate constant which is non-zero and corresponds to the folding rate constant inside the chaperone. From this finding, the authors further explore the mechanism of chaperone-mediated folding, using additional experiments in synergy with site directed mutagenesis.

It was a real pleasure to read this work, which is carefully executed and nicely described. The quality of the data is clearly very good (as evident from inspection of the Figures) and the statistical analysis appears solid.

I have no criticism and do recommend publication.

Reviewers' comments:

Reviewer #1 (Remarks to the Author):

In their paper “protein folding while chaperone bound is dependent on weak interactions” Wu et al., used two mutants of the ATP-independent periplasmic chaperone Spy and they describe a new Spy client (substrate), SH3, to further characterize the mechanism by which the chaperone accelerates its client’s refolding to their native state. They measured the kinetics of the client intrinsic fluorescence in the presence of increasing Spy concentrations. The global fitting of the data strongly suggested that, as they previously established for the Im7 substrate, SH3 too is refolding while bound to the chaperone and not after dissociation. They characterized two Spy mutants, H96L and Q100L, that bound the native state of SH3 much tighter than WT Spy, and this reduced the rates of Spy-mediated native refolding. They complemented their study with kinetics of ThT fluorescence of SH3mut Δ 57 fibril formation and showed that the SpyQ100L with the highest affinity for the native state effectively prevented amyloid formation of the SH3 variant. Moreover, excessively tight binding of native SH3 by SpyQ100L seemingly caused a slow unfolding. Overall, the data strongly indicates central the role of weak interactions between Spy and the natively folded terminal intermediate of the folding reaction which is being accelerated by the Spy chaperone. Overexpression showed the biological relevance of the abnormally high-affinity of the Spy Q100L mutant for its substrate(s), which caused severely impaired cell division.

This paper provides important new in-depth insights on the molecular mechanism of a molecular chaperone that does not need ATP hydrolysis to accelerate and increase the yields of native refolding of two chemically pre-unfolded clients. Although some aspects of the Spy mechanism have been described previously with the Im7 client, notably the refolding of the client while still bound to the chaperone, SH3 is a very different client, which has a more complicated folding pathway. Hence, this work provides a much-needed additional example to generalize the Spy chaperone mechanism. Moreover, the two Spy mutants with abnormally high affinities for their clients, confirmed a general principle in enzymology that high affinity for the substrate is not always a corollary to high and effective enzymatic activity, especially when high affinity of a bound late intermediate may retard the clearing of the active site, thereby preventing the iterative processing of other substrates.

Major concern:

The evidence that SH3 folding occurs while bound to the chaperone is convincingly, albeit indirectly inferred from the best fits of the kinetic curves. Other experimental approaches should be undertaken to show more directly that the native refolding of SH3 occurs while bound to Spy and not following dissociation. One obvious approach would be to engineer an SH3 with two bound low molecular weight fluorophores allowing assessment by FRET of the different possible states of the client protein: native, unfolded, misfolded and/or aggregated. This would likely confirmed by single molecule studies that SH3 folding occurs while bound to Spy. Other methods should be considered, such as chemical crosslinking and mass spectrometry, limited proteolysis-coupled to mass spectrometry, BIFC, SPR, etc.

Note: New portions of the manuscript and those that are substantially modified are shown in **yellow highlighting**.

Response:

We are happy that the reviewer finds our data convincing! In particular, we highlight the fact that folding still occurs at very high Spy concentrations is consistent only with refolding occurring while

chaperone bound. As an additional approach, to show folding while bound even more directly, the reviewer suggests that we “engineer an SH3 with two bound low molecular weight fluorophores allowing assessment by FRET of the different possible states of the client protein: native, unfolded, misfolded and/or aggregated”. We have now done almost exactly as he or she requested, except that we have utilized a pair of adjacent existing tryptophans in SH3 as the donor, which enabled us to only engineer in a single FRET acceptor (an introduced cysteine modified with TNB) instead of two. This had several advantages, putting in two labels would have had a greater chance of altering the SH3 folding pathway than the introduction of a single FRET label. In addition, engineering in two small molecule FRET pairs at sufficient yield and purity proved to be technically very problematic. We consulted with Sander Tans, an expert on analysis of chaperone action using single molecule studies <https://www.sandertanslab.nl/>, with the hope that he might be willing to do the suggested single molecule FRET studies, possibly in exchange for authorship on the revised manuscript. Sander made the very valid points that such single molecule FRET studies would necessarily involve also labelling Spy and that analysis of data coming from such 3 way FRET studies would be technically extremely challenging, and thus declined our offer. He also noted that we would have no technical way of initiating folding (via denaturant dilution) and Spy binding in single molecule FRET experiments on the ~1 second timescale that SH3 refolds in. He suggested that bulk FRET studies would be more informative, we got similar advice from another FRET expert we consulted, Daniel Raleigh, so we proceeded in that direction. In the new experiments, now incorporated into the manuscript, we monitored the refolding of TNB-labeled SH3 in the absence and in the presence of Spy and used that data to calculate donor-acceptor difference between the two fluorophores during the refolding and binding processes. Our new FRET data show that the unfolded conformation of SH3 immediately after dilution from denaturant is more compact than the SH3 conformation in high concentrations of denaturant. Furthermore, Spy binding to this denaturant-diluted conformation actually causes a slight further compaction of this state. However, the unfolded conformation is still able to fold all the way into its native state based on a comparison between our measured FRET distance and the expected distance of from the SH3 structure. This changes with SpyQ100L, though, as this mutant stabilizes and traps the SH3 in a non-native conformation. We thank the reviewer for suggesting we perform these FRET studies as we think they definitely strengthen the manuscript!

Minor concern:

The authors should try to discuss more in depth the implications of their results for the chaperone mechanism of action on its clients. For example, on the one hand, the refolding kinetics strongly suggested that the pre-unfolded SH3 was led to rapidly reach its native state while bound to the chaperone. Thus, in principle, if the stoichiometry of the reaction was set to undergo a single turn over, a slower dissociation rate of the natively refolded client from the high-affinity chaperone mutants, should not affect the rate and the yield of Spy-mediated refolding of SH3 to the native state, as observed. If several turnovers of more than a single client per Spy active site, were necessary to produce the observed refolding rates, this could explain the lower values observed with the high affinity Spy mutants. Slower active site clearance would impair iterative processing of the intermediate species. Consequently, the paper would gain from showing kinetics of SH3 refolding to the native state under conditions where the unfolded SH3 would be in a large molar excess over Spy.

The experiment the reviewer suggests ie “showing kinetics of SH3 refolding to the native state under conditions where the unfolded SH3 would be in a large molar excess over Spy” has historically given interpretable data for chaperones such as GroEL using substrates that are stringent substrates, ie they

only fold in the presence of the chaperone (refs). Figure 5a in Todd et al PNAS 93: 4030-4035 (1996) seems like a good example of what the reviewer is thinking of. In this case rubisco on its own is unable to fold, but even a 1 nM concentration of GroEL is able to cause ~10 nM of rubisco to refold, supporting the iterative-annealing (i.e., catalytic) mechanism for GroEL. An important difference here, though, is that Rubisco either aggregates or does not fold spontaneously. SH3 always folds spontaneously, and at a rate that is faster than the Spy-bound rate. Because SH3 in isolation folds faster than the Spy-bound rate, I would expect SH3 to just fold at its unbound refolding rate in the presence of sub-stoichiometric Spy concentrations. Thus, for substrates like SH3 that fold in both the presence and absence of the chaperone we do not expect to see any clear effect for experiments done under conditions of high substrate to chaperone ratio. We did go ahead and do the experiment the reviewer suggested and this is exactly what we saw.

Another central question of this system is what is the precise state of the pre-denatured substrate when, following dilution of the denaturant, it first interacts with the Spy? Is it mostly unfolded? Is it mostly already misfolded (because of a rapid hydrophobic collapse) or would it be aggregated?

Response:

We measured SH3's refolding kinetics at several SH3 concentrations (0.5 μ M-8.7 μ M) to determine whether or not the protein aggregated upon dilution from denaturant. All SH3 concentrations produced the same observed rate constant (Supplementary Figure 3a), and the amplitude of the refolding traces increased linearly with the SH3 concentration (Supplementary Figure 3b), indicating that the denaturant-diluted SH3 had not aggregated. If the protein irreversibly aggregated after dilution, we would have expected the relative refolding amplitudes to have gotten smaller with increasing SH3 concentrations because the rate of aggregation increases with increasing protein concentration. Even if the protein reversibly aggregated, the observed rate constant should also change with SH3 concentration. We did not observe either of these effects when changing the SH3 concentration, giving us confidence that SH3 did not aggregate in our experiments.

To tackle the question of the precise state of denaturant-diluted SH3 when it interacts with Spy, we used a TNB-labeled SH3 to measure the distance between SH3's tryptophan donor and the TNB acceptor for the kinetic traces at time zero (immediately after dilution from denaturant). The donor-acceptor distance of SH3 immediately after dilution from denaturant was 21.4 Å, compared with a 26 Å distance in the presence of 9.5 M urea (Fig. 2a). We interpret this as indicating that the unfolded state of SH3 immediately after dilution from denaturant has undergone some level of rapid hydrophobic collapse. Furthermore, our FRET kinetic data (Fig. 2a-b) showed that at time zero during SH3 refolding, the donor-acceptor distance was lower in the presence of Spy (~19.3 Å) than in its absence (~21.4 Å). This data suggests that Spy further compacts the conformation of unfolded SH3 immediately after dilution from denaturant. This result is consistent with the previous observation by NMR that Spy compacts the unfolded state of Im7 (Salmon, L., JACS, 2016; He, L., Science Advances, 2016)

The finding that SH3mut Delta57 fibril formation was prevented by SpyQ100L strongly suggests that even WT Spy could bind partially structured misfolding and aggregating intermediates, and not only as generally presumed, to unfolded entrant substrates, slowly turning thereafter into late bound natively refolded intermediates.

Examination of Fig 6 suggests that the SpyQ100L mutant binds the native SH3 so strongly that following initial binding, Spy could be causing slow structural changes in the bound protein that could be interpreted as partial unfolding. This would suggest that following initial binding of the denatured client

to WT Spy, the latter would exert an unfolding action on the substrate, as shown in several instances to be the case with the ATP-dependent chaperones GroEL, DnaK and Hsp104. Could the Spy-mediated unfolding of misfolded clients explain the observed slow exponential decrease of fluorescence in Fig 6? In contrast, if the entrant substrate was to be predominantly unfolded at the time of binding, the slow exponential decrease of fluorescence could result from chaperone-mediated compaction, since further unfolding of an already unfolded polypeptide is a less sound possibility.

Here again, knowing the precise state of the pre-denatured substrate and its fate during initial binding, and during the subsequent slow exponential phase and until it has finally reached its native state, while bound, could/should be addressed by FRET with double-labelled SH3 and/or Im7.

Response:

We attribute the slow exponential decrease in fluorescence observed with SpyQ100L to Spy-mediated unfolding of SH3. Because SH3 is fully in the native state at the beginning of these stopped-flow experiments, the exponential decrease is more consistent with unfolding than binding and compaction of unfolded molecules. The percentage of the population of SH3 molecules which are unfolded should be determined by the affinity of chaperone variants to unfolded molecules. Based on our kinetic models we were able to calculate the relative populations of unfolded and native SH3 that are predicted to be present in the presence of excess of either wild type Spy, SpyH96L or SpyQ100L (supplementary Figure 8 in current version of the manuscript). We calculate that ~30% of SH3 molecules should be unfolded in the presence of SpyQ100L, whereas only ~3% of SH3 is in the unfolded state in the presence of wild-type Spy, due to SpyQ100L's much stronger binding affinity for unfolded SH3 compared with wild type Spy. To experimentally verify that SH3 undergoes unfolding in the presence of SpyQ100L, we conducted stopped-flow FRET kinetics by mixing TNB-labeled native SH3 with SpyQ100L. The D-A distance increased by 1.3 angstroms in the presence of SpyQ100L (Fig. 5c) with kinetics that show the same rate constant as observed in similar stopped-flow experiments with SpyQ100L where we followed the tryptophan fluorescence of unlabeled SH3 (Figure 5a). In contrast, there was very subtle effect on the D-A distance when SpyWT and SpyH96L binds to TNB-labeled native SH3. This data agrees with the notion that SpyQ100L increases the population of unfolded SH3 due to their strong binding affinity for the non-native client.

In the discussion, trigger factor and Hsp33 are mentioned and the authors discuss about a possible ancestral mechanism of chaperone mediated folding without ATP that would require weak chaperone-interactions. This mechanism would have appeared in evolution before the complex GroEL/GroES mechanism, which can use the energy of ATP hydrolysis to unfold and possibly also to drive the dissociation of over-sticky clients. Whereas, this is probably true in most of the cases, very early reports from the Fersht lab showed that some proteins could be folded without ATP hydrolysis, solely by single or seven fused GroEL apical domains (called mini-chaperones). Moreover, small HSPs, which like Spy are not ATPase chaperones, should also be mentioned in this discussion. They have been considered by many to be the most ancestral of all chaperones as they are present in all organisms, plants, fungi metazoans, eubacteria as well as the most primitive forms of Archea. The last universal common ancestor (LUCA) was predicted to contain only sHSPs as Hsp60 chaperonins to control proteostasis. LUCA was missing Hsp70, Hsp90, Hsp100 and notably also Spy. sHSPs likely work by way of maintaining weak interactions with their clients but this remains to be experimentally established.

Response:

We have now referred to the mini-chaperone work from the Fersht lab in the discussion. We have also added a paragraph comparing and contrasting the action of sHSPs and Spy to the discussion.

Reviewer #2 (Remarks to the Author):

The manuscript by Wu et al. exploits stopped-flow kinetics experiments to investigate the interaction between the ATP-independent periplasmic chaperone Spy and a client SH3 domain. Based upon their data, the authors propose that the stronger the interaction is, the slower is the client's refolding.

In general, I found that the topic is of interest, especially because the idea of a chaperone mediating refolding while bound to its client, rather than after its release, is novel. The results are clear.

Response: We thank the reviewer for his or her kind comments!

However, I found that the manuscript lacks the details and the resolution that would be necessary to support authors' conclusions in a high impact journal. Apart from the ITC in the first section, the manuscript takes advantage of but one experimental probe, tryptophan intrinsic fluorescence, which can be monitored only with cut-off filters in stopped flow experiments. Second, I have personally employed kinetic analyses a lot and I know it is a very useful tool, yet it never provides direct evidence, it just tells which is the most likely among a set of models examined.

Response: Reviewer 1 made similar points, that kinetic analysis, though useful and convincing are necessarily indirect and suggested that we undertake FRET experiments to provide more direct evidence to support our conclusions. We think the new FRET experiments we have now included enhance the manuscript as they provide distance-based conformational information about the different states that are observed in our tryptophan fluorescence kinetics.

More specific comments:

I think the authors should describe more clearly their probe. How many tryptophans does SH3 have? Where are these located? I guess Spy does not have any tryptophan or this would affect data interpretation. This should be briefly discussed.

Response:

SH3 has two consecutive tryptophans in its sequence at positions 36 and 37. They form part of the hydrophobic core of native SH3 and form the donor for the new FRET experiments. Spy does not have any tryptophans. We have clarified these points in the manuscript.

I have several doubts about the ThT-amyloid section and it is not very clear to me how these results fit into the rest of the manuscript. The authors take advantage of a variant of SH3, known to be aggregation prone. Spy inhibits aggregation and the authors conclude that these results are consistent with Spy providing a folding-friendly environment. However, no one knows how this inhibition is

achieved and the variant employed here was reported to be a three-state folder in Neudecker et al. 2012, with a poorly populated partially folded aggregation prone state. Most likely, Spy binds the folded state (something the authors show for Q100L and H96L), thus shifting the equilibrium and inhibiting population of the aggregation-prone intermediate. What one should investigate here is whether, starting from the unfolded state, Spy is able to suppress the SH3 amyloidogenic intermediate. However, the rest of the manuscript investigates a two-state folder and thus it is not very clear to me what information these experiments provide, with respect to the manuscript main message.

Response:

We thought that the results of ThT fluorescence experiment were interesting because they indicate that Spy can protect proteins from aggregation while they refold. However, we agree with the reviewer's point that it requires more evidence to link this result to the concept of this chaperone's mechanism. Therefore, we decided to remove this amyloid data from the manuscript and plan on eventually submitting it for publication elsewhere.

. The authors claim that Spy Q100L and Spy H96L unfold their clients based upon the decrease in fluorescence that was observed following binding. How can the authors rule out a structural rearrangement, different from unfolding and involving a decrease in fluorescence? Indeed, tryptophans get commonly exposed to the solvent during unfolding and this involves red-shift and an increase in fluorescence above 320 nm, even though this is not a strict rule. The authors point out that the unfolded state of SH3 is less fluorescent than the native state, but I guess this was observed at high [denaturant]. The authors should extrapolate the fluorescence signal emitted by the U state at the denaturant concentration at which they did the binding to Q100L/H96L and determine whether such signal is compatible with that observed at the end of the slow phase.

Response:

We have tried to determine the fluorescence of unfolded SH3 under our experimental conditions (0.83 M urea) by linearly extrapolating the fluorescence of denatured Fyn-SH3 at high urea concentration. However, Fyn SH3 can only be fully unfolded at urea concentrations above 8.5 M; the tiny range of urea (8.5M to 9.5M) that fully denatures SH3 makes it very challenging to accurately determine the fluorescence of unfolded SH3 at 0.83 M urea from a linear extrapolation. As a workaround, we used global fitting of our urea-dependent refolding data to determine the actual fluorescence for the unfolded state and folded state under our experimental conditions. This global fitting procedure takes into account the urea-dependent change in fluorescence of the unfolded and native states throughout all of our data points (0.83 M – 9.5 M urea). From this analysis, the fluorescence of unfolded state can be described by $0.404+0.016*[\text{urea}]$ while the one of folded state can be described by $0.985+0.027*[\text{urea}]$. These formulas are based on the assumption that the fluorescence scales linearly with the concentration of urea. Under our experimental conditions, which contain 0.83 M urea, the fluorescence of unfolded state is 0.42 a.u./uM while the fluorescence of native state is 1.01 a.u./uM. We note that our new FRET experiments indicate that the unfolded conformation of SH3 immediately after dilution from denaturant during our refolding experiments is more compact than the unfolded SH3 conformation in the presence of 9.5 M urea (21.5 Å vs 26 Å donor-acceptor distance). Because of this change in conformation, there is no reason to assume that the tryptophan fluorescence of the denaturant-diluted state should be identical to the unfolded state in high denaturant concentration.

We acknowledge this reviewer's point that we cannot definitively rule out the possibility that the observed decrease in fluorescence could be due to a structural rearrangement other than partial unfolding of SH3. However, the combined results from several pieces of evidence are most consistent

with a change in the ratio of unfolded/native SH3 upon binding of SpyQ100L/H96L. We'll restrict the following explanation to the SpyQ100L data, but the same logic applies to SpyH96L. First, as noted by the reviewer, the fluorescence of native SH3 decreases after binding to SpyQ100L. From Figure 1a of our revised manuscript, the fluorescence of Spy-bound unfolded SH3 (fluorescence at zero seconds immediately after dilution from urea) is ~2.5 and the fluorescence of Spy-bound native SH3 (at the end of the reaction) is ~4. The decrease in native SH3 fluorescence upon binding SpyQ100L from ~4 to ~3.6 in Figure 5a is therefore consistent with a change in the population of native SH3 from 0% unfolded/100% native to ~30% unfolded/70% native based on the new kinetic simulations that we've included in the revised manuscript (Supplementary Fig. 8). Second, the kobs measured when SpyQ100L binds native SH3 is identical to the kobs measured when unfolded SH3 refolds while bound to SpyQ100L. This suggests that both experiments are monitoring the same conformational transition, but from different directions – from unfolded to native SH3 in the refolding experiment and from native to partially unfolded SH3 in the binding experiment. Third, the intramolecular FRET experiments included in the revised manuscript reveal that the native SH3 converts from a more compact conformation (~17 Å donor-acceptor distance) to a more extended conformation (~19 Å donor-acceptor distance) during the slow transition after binding to SpyQ100L. Notably, binding of SpyWT to native SH3 does not cause a change in the conformation of SH3, in correspondence with the abnormally strong binding affinity of SpyQ100L for unfolded SH3 when compared with SpyWT. We have clarified these points in the revised manuscript.

Like the amyloid section, the bacterial fitness section reads a little out of place in this manuscript. If it is intended to suggest that the folding-while-bound capability of Spy enhances fitness, this conclusion is clearly far-fetched, and in fact the authors do not state that. If, instead, it aims to investigate how chaperones affect morphology or other processes, it sounds off topic.

Response:

Though slightly tangential to the main approaches of the manuscript, the bacterial fitness section provides evidence for the in vivo importance of our work, and thus would like to keep it in. Our kinetic data support the notion that a folding-while-bound mechanism requires weak chaperone-client interactions, weak enough to allow sufficient mobility enabling bound clients to fold. We further showed that SpyQ100L can unfold the native protein due to its strong binding affinity for the unfolded state. These observations led us to think that Spy requires weak interactions with their clients in order to function without negatively impacting the fitness of E. coli. Consistent with this interpretation, overexpressing SpyQ100L does negatively impact bacterial fitness as compared to overexpressing wild-type Spy. Unlike most other stress-regulated chaperones, which usually can undergo conformational rearrangements that regulate their binding affinity with their clients, Spy does not have this stress-induced regulation, nor does Spy have the possibility of ATP regulated client affinity. Therefore, we argue that Spy requires weak binding affinity for its clients. That increasing client binding affinity has a deleterious effect on cell growth supports these conclusions, so we would like to keep these in vivo results in.

. All the experimental values in the text come without experimental errors.

Response:

We have now put experimental errors into the measurement in the text, Missing more details are reported in table 1.

Reviewer #3 (Remarks to the Author):

This is a very nice work where the authors demonstrate that a recently discovered client of Spy (an ATP independent chaperone) can fold while bound to the chaperone.

Thanks to a straightforward analysis of the kinetic data, the authors clearly demonstrate that, challenging the SH3 domain with increasing concentrations of Spy, leads to an observed rate constant which is non-zero and corresponds to the folding rate constant inside the chaperone. From this finding, the authors further explore the mechanism of chaperone-mediate folding, using additional experiments in synergy with site directed mutagenesis.

It was a real pleasure to read this work, which is carefully executed and nicely described. The quality of the data is clearly very good (as evident from inspection of the Figures) and the statistical analysis appears solid.

I have no criticism and do recommend publication.

Response: We thank the reviewer for his or her kind comments!

REVIEWERS' COMMENTS:

Reviewer #1 (Remarks to the Author):

The authors have now well addressed, to my full satisfaction, a major concern of mine, which happened to be a major concern of an referee. Other minor points have also been well addressed. I therefore have no further suggestions to improve this important manuscript, which can be published as is.

Pierre Goloubinoff

Reviewer #2 (Remarks to the Author):

I think the authors made a good effort to improve their results.

Authors performed FRET experiments, and as a consequence the manuscript has surely improved. However, there is one thing I do not completely understand. Why did the authors perform FRET intramolecularly? Such an approach is reasonable if one sets up the model to get insight into the refolding mechanism. If, instead, the goal is to obtain direct evidence of the "folding-while-bound" model (I thought this was the case), I would have found it more reasonable to label Spy somewhere near the binding site and study intermolecular FRET. In this model, SH3 tryptophans are exposed in U; consequently, tryptophan fluorescence is expected to increase and FRET efficiency is expected to decrease upon refolding, due to intermolecular FRET occurring more efficiently in U than in F. This holds true only if U and F are bound to Spy. I know I did not specifically request this experiment in my first report yet it seems to me that the authors missed an opportunity here. Can the authors at least comment on this? Am I missing something?

Reviewer #1 (Remarks to the Author):

The authors have now well addressed, to my full satisfaction, a major concern of mine, which happened to be a major concern of an referee. Other minor points have also been well addressed. I therefore have no further suggestions to improve this important manuscript, which can be published as is.

Pierre Goloubinoff

Response:

We thank Pierre!

Reviewer #2 (Remarks to the Author):

I think the authors made a good effort to improve their results.

Authors performed FRET exps, and as a consequence the manuscript has surely improved. However, there is one think I do not completely understand. Why did the authors performed FRET intramolecularly? Such approach is reasonable if one sets up the model to get insight into the refolding mechanism. If, instead, the goal is to obtain direct evidence of the "folding-while-bound" model (I thought this was the case), I would have found more reasonable to label Spy somewhere near the binding site and study intermolecular FRET. In this model, SH3 tryptophans are exposed in U; consequently, tryptophan fluorescence is expected to increase and FRET efficiency is expected to decrease upon refolding, due to intermolecular FRET occurring more efficiently in U than in F. This holds true only if U and F are bound to Spy. I know I did not specifically requested this experiment in my first report yet it seems to me that authors missed an opportunity here. Can the authors at least comment on this? Am I missing something?

Response:

Interpreting intermolecular bulk FRET between a two flexible partners like the chaperone Spy and the substrate SH3 would be in our view, and according to the FRET experts we consulted with, extremely complicated because the readout for the FRET experiments are sensitive to both folding and binding. In part due to these types of problems, as far as we can tell, there are no published intermolecular bulk FRET studies from any protein chaperone-substrate pair. Thus, if the reviewer had explicitly asked for intermolecular bulk FRET we would have respectfully declined. Single molecule intermolecular FRET is more interpretable but technically extremely challenging and we declined to do these studies for the reasons we previously listed.

We thought instead reviewer #2 was instead asking for intramolecular FRET experiments, which are interpretable. The intramolecular bulk FRET experiments we did, allowed us to monitor the structural changes that take place during protein folding,

addressed a concern from Reviewer #2 that that the decrease in fluorescence upon mixing native SH3 with SpyQ100L might not be due to protein unfolding but some conformational changes in SH3 potentially induced by the binding of SpyQ100L, simultaneously addressed a number of comments from reviewer #1 and significantly strengthened the paper.